# Testing the effectiveness of alcohol health warning label formats: An online experimental study with Australian adult drinkers

Emily Brennan[1]*, Kimberley Dunstone[1], Amanda Vittiglia[1,¤a], Sam Mancuso[1,¤b], Sarah Durkin[1], Michael D. Slater[2], Janet Hoek[3], Simone Pettigrew[4], Melanie Wakefield[1]

1 Centre for Behavioural Research in Cancer, Cancer Council Victoria, Melbourne, Victoria, Australia,
2 School of Communication, The Ohio State University, Columbus, Ohio, United States of America,
3 Department of Public Health, University of Otago, Wellington, New Zealand, 4 The George Institute for Global Health, University of New South Wales, Sydney, New South Wales, Australia

¤a Current address: Australian Institute of Family Studies, Melbourne, Victoria, Australia
¤b Current address: Research and Evaluation Branch, National Disability Insurance Agency, Richmond, Victoria, Australia
* emily.brennan@cancervic.org.au

## Abstract

Health warning labels (HWLs) on alcohol containers may help reduce population-level alcohol consumption. However, few studies have examined the most effective formats for alcohol HWLs. This study tested the effects of three different types of alcohol HWLs. In an online experiment, $N$ = 1,755 Australian adult drinkers were randomly assigned to one of five conditions: (a) No HWL control; (b) DrinkWise control (industry-developed labels currently on some containers); (c) Text-Only HWLs; (d) Text + Pictogram HWLs; or (e) Text + Photograph HWLs. In the three intervention conditions, participants were exposed to eight HWLs, each depicting a different long-term harm. Exposure occurred during an initial session, and repeatedly over the subsequent eight days. Differences between conditions were assessed immediately following initial exposure and at nine-day follow-up. Compared to participants in the No HWL control, participants exposed to Text + Pictogram HWLs were more likely to have intentions to avoid drinking alcohol completely in the next month (post-exposure) and intentions to drink less alcohol in the next week (follow-up), and participants in all three intervention conditions reported stronger negative emotional arousal (follow-up) and weaker positive emotional arousal (follow-up). Compared to participants in the DrinkWise control, those exposed to Text + Pictogram HWLs had stronger intentions to drink less alcohol in the next week and intentions to avoid drinking alcohol completely in the next month (follow-up), participants in the Text + Photograph condition reported significantly weaker positive emotional arousal, and all three intervention conditions resulted in stronger negative emotional arousal. There would likely be benefits to public health if any of the three types of intervention HWLs were implemented. However, there is some evidence that Text + Pictogram HWLs should be recommended over Text-Only or Text + Photograph HWLs, given they were the only HWLs to increase intentions to drink less.

**Data Availability Statement:** The data analyzed for this study will be available in the openICPSR repository.

**Funding:** The study was funded by an Australian National Health and Medical Research Council (https://www.nhmrc.gov.au/) Project Grant [#1129002] awarded to MW, EB, SD, SP, JH, and MDS. The funders had no role in study design, data collection and analysis, the decision to publish, or preparation of the manuscript.

**Competing interests:** I have read the journal's policy and the authors of this manuscript have the following competing interests: EB, KD, SD and MW are employed by a non-profit organisation that conducts research, public health interventions and advocacy aimed at reducing alcohol-related health harms in the community, especially those pertaining to cancer. EB, SP and MW have received other NHMRC grants on alcohol harm communication. MDS has current grant support for alcohol-related research from the NIH National Institute on Drug Abuse and has previously received grants from the National Institute on Alcohol Abuse and Alcoholism, National Cancer Institute, and the Robert Woods Johnson Foundation. EB and SP are members of the Expert Reference Group for the Alcohol. Think Again campaign. JH has provided advice to Alcohol Healthwatch, a New Zealand non-profit organisation that conducts advocacy to reduce alcohol-related health harms in the community. This does not alter our adherence to PLOS ONE policies on sharing data and materials.

# Introduction

Alcohol consumption is causally linked to over 60 health harms, including liver disease, heart disease, stroke, and at least seven types of cancer [1–3]. Introducing health warning labels (HWLs) on alcohol containers could educate consumers about alcohol-caused harms and reduce alcohol consumption [4]. As of 2018, 47 World Health Organization member countries had mandated alcohol HWLs [5]. Many of these HWLs have been characterised as having weak written messages [6] and most lack images [7], have low noticeability [8], and feature harms that allow most drinkers to self-exempt (e.g., drinking when pregnant) [4]. To date, HWLs that warn about the link between alcohol and cancer have been mandated in South Korea (although industry can select which one of three HWLs is placed on the container and only one of the three mentions cancer) [9] and in Ireland [10], although industry interference there has reportedly delayed both the passage of the bill and the introduction of the HWLs [11–13].

In Australia, DrinkWise Australia, an alcohol industry-funded Social Aspects/Public Relations Organisation has produced labels since 2011 [14–16] that have been criticised for vague wording and weak images [17–20]. In 2020, only 31% of Australians recalled seeing these labels [21], likely due to their limited use by the industry, very small size, location on the back of containers, and use of the same colour palate as product branding [20, 22]. Food Standards Australia New Zealand recently mandated a pregnancy HWL to appear on all alcohol containers [23], with a minimum size requirement and a white background, red and black text and a red and black pictogram that must be implemented before 31 July 2023 [24].

While most Australians are aware of the recommendation to avoid alcohol during pregnancy [25], and of the link between alcohol use and liver cirrhosis and liver cancer [26], awareness of many other alcohol-related harms remains low [25, 26]. Fewer than one-third are aware that alcohol can cause mouth cancer, throat cancer and breast cancer [26]. HWLs therefore have considerable scope to improve population awareness of harms, with the broader goal of limiting alcohol consumption. Australians' support for mandatory alcohol HWLs is reasonably high [27], perhaps due to the long history [28] and widespread acceptance [29] of its tobacco HWLs. Reviews conclude tobacco HWLs increase health knowledge and perceptions of risk, elicit negative emotional responses, increase quitting and prevent youth initiation [30], with those most effective including a full colour image and covering at least 50% of front of the package [31–34]. The effects of tobacco HWLs have been found to wear-out over time, as smokers stop paying attention to them [35–37]. To slow down these wear-out effects, many tobacco HWL policies require a set of HWLs to be implemented and/or for two or more sets of HWLs to be rotated (although more recent research indicates that these measures are not completely effective at preventing wear-out such that frequent introduction of new HWLs is required) [38].

To date, few studies have examined the alcohol HWL elements that may contribute to effectiveness. An Australian qualitative study suggested that large warnings with pictograms or graphic photographs may increase attention, negative emotional responses and understanding of harm messages among young adults [39]. Other research suggests positively framed statements (e.g., 'Reduce your drinking to reduce your risk') perform better than negatively framed statements (e.g., 'Alcohol harms your health') [40], and that warnings referring to specific cancers perform better than general cancer warnings [40, 41]. A study in the United Kingdom (UK) tested image-and-text, image-only, and text-only HWLs that communicated the link between alcohol and either bowel, breast or liver cancer [42]. The HWLs were selected for their potential to elicit negative emotions and reduce the desire to consume alcohol [43]. Compared to no HWL, exposure to HWLs increased the likelihood of drinkers selecting a non-

alcoholic beverage, and HWLs that included an image (with or without text) were more effective than text-only HWLs [42]. In Canada, a real-world quasi-experimental study used systematically-developed HWLs that were large, colourful, and included a text-only cancer warning, a pictogram plus text describing Canada's National Drinking Guidelines and a pictogram plus text educating about the number of standard drinks in a container [44]. These HWLs were placed on alcohol containers sold in the sole government-run liquor store in an intervention location, while usual practice (i.e., long-standing warnings about drinking when pregnant, and about the impact of drinking on operating a car or machinery and general health risks [45]) continued at two liquor stores in the control location. After one month, participants at the intervention site were more likely than those at control sites to recall a HWL about alcohol and cancer, to have knowledge of the link between alcohol and cancer and of the low-risk drinking guidelines and to intend to reduce their alcohol intake [46–48]. Furthermore, implementation of the HWLs significantly reduced alcohol sales at the intervention location [45].

Jurisdictions that are implementing alcohol HWLs for the first time will need to decide whether to implement prominent text-only warnings or pictorial warnings that feature simple stylised pictograms or that feature graphic photographs. Evidence from tobacco control has demonstrated the superior effectiveness of pictorial warning labels over text-only labels [30–33, 49] and the UK study described above found a similar advantage for alcohol HWLs that include an image [42]. The current study aims to build on previous research in several ways. First, we assessed a range of distinct HWL formats: Text-Only, Text-and-Pictogram and Text-and-Photograph. Second, our suite of systematically developed alcohol HWLs covers a range of relatively lesser known alcohol-related long-term health effects. Third, we compared the effectiveness of the three warning formats against two control conditions. The No Warning Label control condition reflects most alcohol containers in Australia which do not carry any warning labels and the DrinkWise control condition reflects the fewer than half of alcohol containers in Australia that carry one of the warnings developed by DrinkWise Australia [20–22]. Fourth, we sought to mimic the real-life scenario whereby consumers are exposed to alcohol containers on a regular basis by supplementing an initial dose of exposure to the HWLs with repeated daily exposure for eight days.

## Methods

### Design and setting

We conducted a between-subjects online experiment with an initial exposure session, measurement of immediate post-exposure outcomes, repeated exposure over the subsequent eight days and measurement of additional outcomes at follow-up. Australian adults were randomised to one of five conditions: (1) No HWL control; (2) DrinkWise control; (3) Text-Only intervention (Text-Only); (4) Text-and-Pictogram intervention (Text + Pictogram); or (5) Text-and-Photograph intervention (Text + Photograph) (Fig 1). In each of the three intervention label conditions, participants were exposed to eight HWLs, each depicting a different alcohol-related harm.

Based on prior unpublished research, we aimed to detect a magnitude of difference of 12 percentage points in greater intentions to limit drinking at follow-up between the intervention and control conditions. Power calculations showed that $n = 364$ participants per condition at follow-up would detect this difference at 90% power, $\alpha = 0.05$, and so we aimed to recruit $n = 520$ per condition ($N = 2,600$ in total) to allow for an expected 30% attrition rate at follow-up (anticipated by the online panel provider).

**Fig 1. Condition allocation and flow of participants through study.**

## Participants

Recruitment and data collection commenced on 12 March 2020. Due to the rapidly changing environment and widespread uncertainty caused by COVID-19, recruitment ceased prematurely on 25 March 2020 and all follow-up data collection was completed by 8 April 2020.

In total, $N = 1,829$ participants were recruited, but 144 dropped out after randomisation (Fig 1), leaving $N = 1,755$ (67.5% of the planned sample size) for analyses of outcomes measured immediately post-exposure and $N = 1,087$ (59.7% of the planned sample size at follow-up) for analyses of follow-up outcomes. We determined the minimum difference in intentions to limit drinking at follow-up that could be detected with 90% power, $\alpha = 0.05$, given the smallest achieved number of participants per condition at follow-up (n = 209 in No WL control and n = 211 in Text + Pictogram; Fig 1). Using the observed proportion for the No WL control condition as the reference proportion (Table 2), we were powered to detect a difference of 15.7 percentage points in intentions to drink less in the next week at follow-up. Proportions in each intervention condition on this outcome (Table 2) indicated that the observed differences were clearly smaller (7.8 percentage points, 8.1 percentage points) or larger (17.2 percentage points) than the range of 12 percentage points (a priori power analyses)– 15.7 percentage points (powered to detect with achieved sample), indicating that the lower-than-planned sample size did

not meaningfully impact our capacity to detect significant differences between control and intervention conditions on this outcome.

All participants were aged 18–69 years and consumed alcohol on average at least weekly during the past year (18 is the minimum legal drinking age in Australia). Quotas were applied to achieve approximately even numbers by gender and proportional quotas for age (18–29, 30–49 and 50–69 years) based on the distribution of weekly drinkers aged 18–69 years in 2019 [50]. Participants were recruited through an online non-probability panel accredited under the International Organization for Standardization's standards for Market, Opinion and Social Research (AS ISO 20252), where participants opt-in to receive email invitations to participate in market research. Survey participants receive points they may accrue and redeem for gift vouchers.

Ethical approval for the study was obtained from Cancer Council Victoria's Institutional Research Review Committee (IER 1609). Participants received an email invitation to participate in the study, clicked through to the study, were provided with information about the broad aims of the study, completed screening questions and then implied consent by clicking through to further questions. This implied consent was collected in place of formal written or verbal consent. This procedure was approved by the Institutional Research Review Committee. The study protocol and analysis plan were pre-registered (http://www.ANZCTR.org.au/ACTRN12620000111976.aspx).

## Procedure

Panel members received an email invitation, clicked through to the study, completed screening questions and then implied consent by clicking through to further questions assessing alcohol consumption, after which they were randomised to condition. Participants were then asked to indicate their preferred type of alcohol from beer, wine or spirits. Exposure to the HWL stimuli was achieved via eight Drink Choice Tasks (DCTs). At the beginning of the DCTs, participants were asked to visualise a scenario where they might drink over the next few days. They were informed that they would be presented with different brands of alcohol. In each DCT they were then asked: "Thinking about the drinking scenario you pictured yourself in, which of these drinks is most appealing to you?", with a "None appeal to me" option.

In each DCT, images of the drinker's preferred alcohol type were presented on three alcohol containers that differed by brand but not by HWL; that is, all three containers featured the same HWL (or none, in the No HWL condition). As participants in the three intervention conditions completed all eight DCTs they were exposed once to each of the eight HWLs (in a random order), while participants in the DrinkWise condition were exposed twice to each of the four DrinkWise labels (in a random order). We used eight different intervention HWLs to achieve a balance between covering a range of health effects and ensuring sufficient exposure to the HWLs without overburdening participants.

The three brands presented in each DCT came from a random selection of eight brands for each alcohol type. Across the eight DCTs, there were 56 possible brand combinations that a given participant could see in each DCT. Therefore, by chance any given participant likely saw a different array of three brands across each of the eight DCTs. S1 Fig provides an example of how brands and HWLs varied across the DCTs and S2 Fig provides an example of how the DCTs appeared to participants.

After completing the eight DCTs, participants reported on intentions to reduce their drinking and answered additional demographic questions. They were then invited to participate in the 8-day repeated exposure task, which required them to open a daily email in which they were exposed to an image of an alcohol container of their preferred type, bearing one of the HWLs relevant to their condition. Each day, they were asked to rate the appearance of this alcohol container on a 5-point scale ranging from very unappealing to very appealing. This

question aimed to ensure that participants at least minimally engaged with the stimuli; responses to these questions were not analysed. Participants could choose to complete the repeated exposure task on some days but not others, although the incentive structure meant they received additional points if they completed at least six of the eight tasks. By allowing variation in the number of repeated exposure tasks completed by each participant (between zero and eight), we aimed to mimic the real-life scenario whereby consumers are exposed to alcohol containers on a regular basis, but with a frequency that varies across consumers and over time (i.e., even a regular drinker will be exposed more frequently on some weeks than others).

The follow-up survey sent to participants on day nine of the study assessed alcohol consumption over the past week, drinking reduction intentions, knowledge gain in relation to alcohol-related health harms, emotional responses to the warning labels, and whether they had shown or mentioned the warning labels to others.

## Stimuli

**Alcohol warning labels.** Intervention conditions: Across the three intervention conditions, the HWL for each of the eight harms featured the same text message: (i) Alcohol is high in kilojoules and so causes **weight gain**; (ii) Drinking alcohol can increase your risk of **heart failure**; (iii) Drinking alcohol increases the risk of **bleeding in the brain**; (iv) Alcohol can increase your risk of **cirrhosis** and **liver damage**; (v) Alcohol can increase your risk of **liver cancer**; (vi) Drinking alcohol increases the risk of bowel cancer; (vii) Alcohol increases the risk of **throat cancer**; and (viii) Alcohol increases the risk of **at least 7 types of cancer**. In the Text + Pictogram condition, the text was accompanied by a black-white-and-red pictogram and in the Text + Photograph condition, the text was accompanied by a colour photograph (see S3 Fig for an example of one Text Only, Text + Pictogram and Text + Photograph HWL; all intervention HWLs are available from the authors upon request). Each HWL was positioned on the front centre-bottom of the containers, covering 20%-28% of the front surface height (Fig 2).

DrinkWise control condition: We adapted four DrinkWise Australia labels available in late 2016, all of which communicated the message 'It is safest not to drink while pregnant', with some accompanied by a pictogram of a pregnant woman [22], and others accompanied by the tagline 'Get the facts DrinkWise.org.au' [16]. For the purposes of our study, we positioned the four DrinkWise labels in the same location as the Intervention HWLs (Fig 2). The DrinkWise labels were in a somewhat larger size than they currently appear (covering around 4–8% of the front surface height) and we used a consistent approach to the colour by using dark text and pictograms for lightly coloured alcohol containers and light text and pictograms for dark coloured alcohol containers (compared to the approach used by many alcohol producers, which is to print the DrinkWise label in colours used in the alcohol labelling) (examples of the DrinkWise labels are available from the authors upon request).

**Alcohol container stimuli.** All HWLs were displayed on images of containers for the eight top selling brands for each alcohol type (beer, wine and spirits), selected using Euromonitor International's 2017/2018 alcohol industry reports for Australia [51–53].

## Measures

**Demographic and drinking characteristics.** Demographic characteristics: Participants reported age, gender, highest level of educational attainment and if they were a parent or guardian. Socioeconomic status was determined using participants' postcode and an Index of Relative Socio-Economic Disadvantage [54].

Drinking characteristics: Participants' past 12 month usual pattern of alcohol consumption was measured using the graduated quantity–frequency measure [55–57]. Participants were

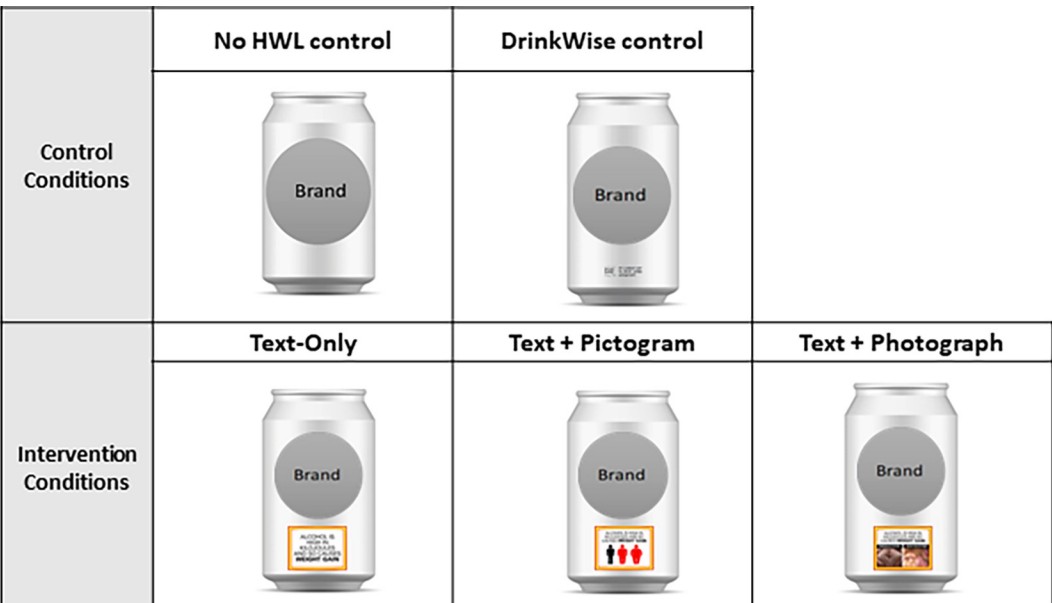

**Fig 2. Appearance of warning labels on one example of an alcohol container (beer).**

provided with a visual guide of the number of standard drinks (one standard drink in Australia = 10g alcohol) in common servings of different alcohol types. Applying the Australian National Health and Medical Research Council's 2009 Low-Risk Drinking Guidelines [58], current at the time of the study, participants were classified as at high risk of long-term harm (LTH) if they consumed > two drinks per day on average and/or at high risk of short-term harm (STH) if they had > four drinks on any occasion at least once a month; we compared those who were at low risk of both LTH and STH with those who were at high risk of either or both LTH and STH. The seven-day follow-back measure was used to record participants' recent pattern of alcohol consumption and calculate the average number of drinks consumed per day in the past week [59].

Participants also indicated the perceived amount of alcohol they currently drink, with 'self-perceived high-risk drinkers' classified as those who responded that "I definitely drink more than I should" or "I probably drink more than I should", and 'self-perceived low-risk drinkers' as those who responded with either "the amount I drink is ok" or "I could drink more than I do".

**Outcomes.** Intentions to reduce drinking in the next week and month: Immediately post exposure, participants indicated on an ordinal scale (1 = "definitely will not", 2 = "probably will not", 3 = "probably will", and 4 = "definitely will") the extent to which "In the next week, will you try to drink less alcohol", and the extent to which they will in the next month, "reduce how often you drink alcohol", "reduce the amount of alcohol you have on each drinking occasion," and "avoid drinking alcohol completely". Responses were dichotomised and classified as 'definitely/probably will not' or 'definitely/probably will'. Since the items "reduce how often you drink alcohol" and "reduce the amount of alcohol you have on each drinking occasion"

were highly correlated (polychoric correlation ρ = .72), they were combined, with those who answered 'definitely/probably will' for either or both statements being classified as 'definitely/ probably will' in the combined variable [60].

Past week alcohol consumption was assessed at follow-up using the seven-day follow-back measure [59].

Intentions to reduce drinking in the next week and month at follow-up were measured using the same questions as those completed immediately post-exposure.

Readiness to change: We used the readiness to change ruler for decreased drinking at follow-up [61]. Participants indicated their readiness to reduce their drinking using an 11-point scale (with the anchor points of 0 = "I never think about drinking less", 3 = "sometimes I think about drinking less", 5 = "I have decided to drink less", 7 = "I am already trying to cut back on my drinking", and 10 = "my drinking has changed, I now drink less than before").

Frequency of thinking about alcohol-related health risks in the past week was adapted from tobacco HWL studies [62] and assessed at follow-up. Participants were asked "In the past week, to what extent, if at all, did you think about the health risks associated with drinking alcohol?". Responses were assessed using a four-point scale (with the option of selecting "don't know") and dichotomised for analysis ('somewhat/a lot' or 'not at all/a little/don't know').

Awareness of alcohol-related health harms: Participants were asked if "Consuming alcohol would": (i) "increase your risk of cancer"; (ii) "increase your risk of liver damage"; and (iii) "increase your risk of heart disease" at follow-up. Given that the DrinkWise labels provided the message that 'It is safest not to drink during pregnancy', participants were also asked if "Consuming alcohol would": (iv) "increase the risk of a woman experiencing pregnancy complications". Responses were assessed on a seven-point scale and dichotomised for analysis ('slightly agree/agree/strongly agree' or 'strongly disagree/disagree/slightly disagree/neither agree nor disagree').

Negative emotional arousal: Negative emotional arousal has previously been used to assess the impact of tobacco warnings, and more recently, alcohol HWLs [42]. At follow-up, participants were asked: "Thinking about the images of alcohol containers I've seen as part of this study, I felt: disgusted / afraid / uncomfortable / worried". Responses were rated on seven-point scales with three anchor points (1 = "not at all", 4 = "moderately", and 7 = "very"). A scale indicating participants' negative emotional arousal was created by taking the mean of the four items (Cronbach's α = .92).

Positive emotional arousal: At follow-up, participants were asked: "Thinking about the images of alcohol containers I've seen as part of this study, I felt: excited / pleased". Responses were rated on seven-point scales with three anchor points (1 = "not at all", 4 = "moderately", and 7 = "very"). A scale indicating participants' positive emotional arousal was created by taking the mean of the two items (Cronbach's α = .85).

Show/mention images to others: At follow-up, participants were asked: (i) "As part of this project, you received emails over the past eight days to view images of different alcoholic drinks. Did you show any of the images to other people?" and (ii) "Did you talk about the images with anyone, even if you didn't show them an image?". Response options were 'yes' and 'no'. Responses were combined so that if the participant answered yes to one or both items, they were coded as 'Yes, shared/talked' versus 'No, did not share/talk'.

## Statistical analysis

Data were analysed using Stata/MP version 16.1. Between-condition differences were examined using logistic regression for binary outcomes, linear regression for continuous outcomes and negative binomial regression for count outcomes.

All models were adjusted for covariates. A baseline variable was used as a covariate in all analyses if it was associated with any outcome variable at either timepoint using a liberal alpha of .10 [63]. As such, we included age, gender, educational attainment, parental status, socioeconomic status, risk of harm, self-perceived risky drinking and total past week alcohol consumption as covariates. We also included preferred alcohol type as a covariate because it was a stratification variable.

For each outcome measured immediately post-exposure, one regression model included covariates and condition. For each outcome measured at follow-up, we also examined whether there was an interaction between condition and the dose of repeat exposure to the HWLs using a second regression model that included covariates, condition, repeated exposure dose and a condition-by-dose interaction term. For the primary analyses, a "high dose" of repeat exposure was defined as having completed at least six of the eight repeat exposure tasks and a "low dose" of repeat exposure was defined as having completed zero to five exposure tasks; however, sensitivity analyses explored whether the same pattern of effects was observed when using different cut-offs. Defining a "high dose" of exposure as comprising six or more tasks recognised that although participants were incentivised to complete all eight repeat exposure tasks, it was unrealistic to expect that most participants would be able to comply with this demand of the study. Setting the cut-point at six or more captured those who completed at least the majority ($\geq$75%) of repeat exposure tasks. Including those who did not complete any repeat exposure tasks in the "low dose" category recognises that in the real world, in any given week, there would be some drinkers who were not exposed to any alcohol HWLs, such that no repeat exposure is part of the expected natural variation.

Pairwise comparisons were performed for outcomes for which there was a significant ($p <$ .05) omnibus test for condition and/or the condition-by-dose interaction. Nine pairwise comparisons were performed with each of the three intervention conditions compared to the No HWL and DrinkWise control conditions, respectively, and to each other, adjusting for multiple comparisons using the Bonferroni-Holm method.

## Results

Table 1 presents the sample characteristics at baseline (pre-exposure). There were no baseline imbalances across conditions.

Of the $N$ = 1,755 participants who completed the baseline survey and initial exposure session, 668 (38.1%) did not complete follow-up, leaving $N$ = 1,087 participants for analyses using follow-up outcomes. The attrition rate was similar across conditions; $\chi^2(4)$ = 1.65, $p$ = .799, but was higher than the expected attrition rate of 30%. As shown in S1 Table, there were no imbalances across conditions in the characteristics of participants retained at follow-up.

Of the $N$ = 1,087 who completed follow-up, 691 (63.6%) had a "high dose" of repeated exposure, and this was similar across conditions: No HWL 62%, DrinkWise 64%, Text-Only 61%, Text + Pictogram 64%, and Text + Photograph 67%.

Table 2 presents summary statistics for each outcome, and results of omnibus tests for whether there was any evidence of between-condition differences and condition-by-dose interactions.

### Immediately post-exposure outcomes

There was a significant main effect of condition for intentions to avoid drinking alcohol completely in the next month (Table 2). Participants in the Text + Pictogram condition were more likely to intend to avoid drinking alcohol completely in the next month than those in the No HWL control (OR = 1.85) (Table 3).

**Table 1. Sample characteristics by condition at pre-exposure and between-condition test results (N = 1,755).**

| | No HWL control | | DrinkWise control | | Text-Only | | Text + Pictogram | | Text + Photograph | | Test |
|---|---|---|---|---|---|---|---|---|---|---|---|
| | (*n* = 350) | | (*n* = 352) | | (*n* = 351) | | (*n* = 347) | | (*n* = 355) | | |
| | *n* | % | *n* | % | *n* | % | *n* | % | *n* | % | *p* |
| **Age** | | | | | | | | | | | .624 |
| 18–29 | 180 | 51.4 | 172 | 48.9 | 180 | 51.3 | 178 | 51.3 | 187 | 52.7 | |
| 30–49 | 103 | 29.4 | 111 | 31.5 | 106 | 30.2 | 88 | 25.4 | 104 | 29.3 | |
| 50–69 | 67 | 19.1 | 69 | 19.6 | 65 | 18.5 | 81 | 23.3 | 64 | 18.0 | |
| *Mean (SD)* | | 36.14 (14.40) | | 36.60 (14.32) | | 36.33 (14.62) | | 37.03 (15.87) | | 35.82 (14.94) | .296 |
| **Gender** | | | | | | | | | | | .568 |
| Male | 157 | 44.9 | 168 | 47.7 | 168 | 47.9 | 168 | 48.4 | 182 | 51.3 | |
| Not male | 193 | 55.1 | 184 | 52.3 | 183 | 52.1 | 179 | 51.6 | 173 | 48.7 | |
| **Educational attainment** | | | | | | | | | | | .869 |
| No tertiary education or other | 81 | 23.1 | 87 | 24.7 | 86 | 24.5 | 91 | 26.2 | 93 | 26.2 | |
| Tertiary education | 269 | 76.9 | 265 | 75.3 | 265 | 75.5 | 256 | 73.8 | 262 | 73.8 | |
| **Socioeconomic status^** | | | | | | | | | | | .548 |
| Low | 110 | 31.4 | 105 | 29.9 | 105 | 29.9 | 106 | 30.5 | 92 | 25.9 | |
| Mid-High | 240 | 68.6 | 246 | 70.1 | 246 | 70.1 | 241 | 69.5 | 263 | 74.1 | |
| **Parental status** | | | | | | | | | | | .101 |
| No | 188 | 53.7 | 204 | 58.0 | 200 | 57.0 | 207 | 59.7 | 226 | 63.7 | |
| Yes | 162 | 46.3 | 148 | 42.0 | 151 | 43.0 | 140 | 40.3 | 129 | 36.3 | |
| **Risk of harm (NHMRC Guidelines)** | | | | | | | | | | | .760 |
| Low risk of LTH and STH | 135 | 38.6 | 126 | 35.8 | 128 | 36.5 | 117 | 33.7 | 126 | 35.5 | |
| High risk of LTH and/or STH | 215 | 61.4 | 226 | 64.2 | 223 | 63.5 | 230 | 66.3 | 229 | 64.5 | |
| **Self-perceived risky drinking** | | | | | | | | | | | .149 |
| Self-perceived low-risk drinker | 202 | 57.7 | 172 | 48.9 | 175 | 49.9 | 185 | 53.3 | 187 | 52.7 | |
| Self-perceived high-risk drinker | 148 | 42.3 | 180 | 51.1 | 176 | 50.1 | 162 | 46.7 | 168 | 47.3 | |
| **Preferred alcohol type†** | | | | | | | | | | | .108 |
| Beer | 132 | 37.7 | 121 | 34.4 | 134 | 38.2 | 122 | 35.2 | 129 | 36.3 | |
| Wine | 123 | 35.1 | 116 | 33.0 | 131 | 37.3 | 129 | 37.2 | 105 | 29.6 | |
| Spirits | 95 | 27.1 | 115 | 32.7 | 86 | 24.5 | 96 | 27.7 | 121 | 34.1 | |
| **Past week alcohol consumption** | | | | | | | | | | | .403 |
| *Mean (SE)* | | 13.76 (0.88) | | 13.70 (0.68) | | 13.71 (0.85) | | 15.72 (0.97) | | 13.76 (0.73) | |

*Note*. HWL = health warning label; SD = standard deviation; SE = standard error; NHMRC = National Health and Medical Research Council; LTH = long-term harm; STH = short-term harm

† Measured at the beginning of the Drink Choice Tasks.

^ Based on national quintiles of socio-economic disadvantage where 'Low' is quintiles 1 and 2 (1–40%) and 'Mid-High' is quintiles 3–5 (41–100%)

**Table 2. Adjusted proportions or means, and omnibus test results for condition or condition-by-dose interaction regression analyses.**

| | N | No HWL control | | DrinkWise control | | Text-Only | | Text + Pictogram | | Text + Photograph | | Test | |
|---|---|---|---|---|---|---|---|---|---|---|---|---|---|
| | | %/M | SE | %/M | SE | %/M | SE | %/M | SE | %/M | SE | p | $p_{int}$‡ |
| **Immediately Post-Exposure** | | | | | | | | | | | | | |
| Intentions to drink less in the next week | 1755 | 39.1 | 2.9 | 35.7 | 2.7 | 39.1 | 2.9 | 45.0 | 3.0 | 43.9 | 2.9 | .079 | |
| Intentions to reduce how often and/or how much consumed per occasion in the next month | 1755 | 51.2 | 3.0 | 52.4 | 3.0 | 55.1 | 3.0 | 58.8 | 3.0 | 59.1 | 2.9 | .148 | |
| Intentions to avoid drinking alcohol completely in the next month | 1755 | 13.3 | 1.9 | 15.7 | 2.1 | 16.0 | 2.1 | 22.1 | 2.5 | 20.1 | 2.4 | **.013** | |
| **Follow-Up** | | | | | | | | | | | | | |
| Past week alcohol consumption (mean) | 1087 | 9.4 | 0.5 | 10.9 | 0.5 | 9.8 | 0.5 | 9.6 | 0.5 | 10.2 | 0.6 | .201 | .338 |
| Intentions to drink less in the next week | 1087 | 40.3 | 3.7 | 43.5 | 3.7 | 48.4 | 3.7 | 57.5 | 3.7 | 48.1 | 3.8 | **.008** | .471 |
| Intentions to reduce how often and/or how much consumed per occasion in the next month | 1087 | 50.2 | 3.9 | 53.3 | 3.7 | 52.8 | 3.8 | 63.1 | 3.8 | 55.8 | 3.8 | .124 | .350 |
| Intentions to avoid drinking alcohol completely in the next month | 1087 | 13.6 | 2.5 | 8.6 | 2.0 | 15.9 | 2.7 | 19.6 | 3.2 | 17.2 | 2.8 | **.016** | .413 |
| Readiness to change (mean) | 1087 | 4.4 | 0.2 | 4.5 | 0.2 | 4.4 | 0.2 | 4.9 | 0.2 | 4.7 | 0.2 | .261 | .261 |
| Frequency of thinking about alcohol-related health risks in the past week | 1087 | 27.0 | 3.3 | 29.8 | 3.4 | 29.5 | 3.3 | 36.2 | 3.7 | 28.1 | 3.3 | .294 | .804 |
| *Awareness of alcohol-related harms* | | | | | | | | | | | | | |
| Increase your risk of cancer | 1087 | 62.8 | 3.7 | 58.0 | 3.6 | 68.2 | 3.5 | 69.1 | 3.5 | 69.9 | 3.3 | **.048** | .976 |
| Increase your risk of liver damage | 1087 | 90.0 | 2.2 | 89.6 | 2.3 | 88.4 | 2.3 | 91.5 | 1.9 | 92.3 | 1.8 | .574 | .790 |
| Increase your risk of heart disease | 1087 | 81.9 | 2.8 | 76.2 | 3.2 | 80.6 | 2.8 | 83.5 | 2.6 | 80.9 | 2.8 | .388 | .746 |
| Increase the risk of pregnancy complications | 1087 | 84.5 | 2.8 | 84.8 | 2.5 | 83.4 | 2.7 | 83.1 | 2.9 | 78.0 | 3.1 | .338 | .959 |
| Negative emotional arousal (mean)† | 1087 | 1.8 | 0.1 | 1.8 | 0.1 | 2.7 | 0.1 | 2.9 | 0.1 | 3.1 | 0.1 | < **.001** | **.010** |
| Positive emotional arousal (mean) † | 1087 | 3.3 | 0.1 | 3.1 | 0.1 | 2.7 | 0.1 | 2.9 | 0.1 | 2.5 | 0.1 | < **.001** | .877 |
| Show/talk about images with others | 1087 | 17.8 | 2.9 | 12.8 | 2.2 | 19.8 | 3.0 | 20.3 | 3.0 | 17.6 | 2.8 | .237 | .151 |

*Note.* HWL = health warning label; M = mean; SE = standard error

‡ = *p*-value for condition × dose interaction; bolded p-values are significant at p < .05 and

† = HC3 heteroskedasticity-consistent standard error estimator used.

## Follow-up outcomes

There was a significant main effect of condition for intentions to drink less alcohol in the next week (Table 2). Participants in the Text + Pictogram condition were more likely to intend to drink less than those in the No HWL control (OR = 2.00) and DrinkWise control (OR = 1.76) (Table 3).

There was also a significant main effect for intentions to avoid drinking completely in the next month (Table 2), with participants in the Text + Pictogram condition more likely to intend to avoid drinking than those in the DrinkWise control (OR = 2.59) (Table 3).

There was a significant main effect for awareness of increased cancer risk (Table 2), but no significant pairwise comparisons after adjustment for multiple comparisons (Table 3).

There was a significant main effect for negative emotional arousal (Table 2), with participants in the three intervention conditions each reporting higher negative emotional arousal than those in the No HWL and DrinkWise control conditions (Table 4). In addition, participants in the Text + Photograph condition reported higher negative emotional arousal than those in Text-Only (Table 4). There was also a significant condition-by-dose interaction for negative emotional arousal. Fig 3 shows that in the two control conditions negative emotional arousal scores were higher among those with a low dose of repeated exposure (mean = 2.15 in

**Table 3. Pairwise comparisons for intention and knowledge outcomes for which there was a significant omnibus test.**

| | Intentions to avoid drinking alcohol completely in the next month (Immediately Post-Exposure) | | | Intentions to drink less in the next week (Follow-up) | | | Intentions to avoid drinking alcohol completely in the next month (Follow-up) | | | Awareness of alcohol-related harms: Increase your risk of cancer (Follow-up) | | |
|---|---|---|---|---|---|---|---|---|---|---|---|---|
| | OR | (95% CI) | *p* | OR | (95% CI) | *p* | OR | (95% CI) | *p* | OR | (95% CI) | *p* |
| **No Health Warning Label vs** | | | | | | | | | | | | |
| Text-Only | 1.25 | (0.83, 1.87) | .840 | 1.39 | (0.94, 2.05) | .480 | 1.21 | (0.72, 2.02) | 1 | 1.27 | (0.84, 1.91) | 1 |
| Text + Pictogram | 1.85 | (1.25, 2.74) | **.018** | 2.00 | (1.35, 2.98) | **.009** | 1.56 | (0.94, 2.58) | .516 | 1.33 | (0.87, 2.01) | .920 |
| Text + Photograph | 1.65 | (1.11, 2.45) | .112 | 1.37 | (0.92, 2.04) | .472 | 1.32 | (0.79, 2.21) | 1 | 1.37 | (0.92, 2.06) | .750 |
| **DrinkWise vs** | | | | | | | | | | | | |
| Text-Only | 1.02 | (0.69, 1.51) | .902 | 1.22 | (0.83, 1.78) | .921 | 2.01 | (1.14, 3.55) | .112 | 1.55 | (1.04, 2.31) | .224 |
| Text + Pictogram | 1.52 | (1.04, 2.22) | .203 | 1.76 | (1.19, 2.59) | **.032** | 2.59 | (1.47, 4.56) | **.009** | 1.62 | (1.08, 2.43) | .160 |
| Text + Photograph | 1.35 | (0.92, 1.98) | .625 | 1.20 | (0.82, 1.77) | .692 | 2.20 | (1.24, 3.91) | .056 | 1.68 | (1.13, 2.48) | .090 |
| **Text-Only vs** | | | | | | | | | | | | |
| Text + Pictogram | 1.48 | (1.02, 2.15) | .222 | 1.44 | (0.98, 2.11) | .366 | 1.29 | (0.79, 2.09) | 1 | 1.05 | (0.69, 1.60) | 1 |
| Text + Photograph | 1.32 | (0.90, 1.92) | .604 | 0.99 | (0.67, 1.44) | .945 | 1.10 | (0.67, 1.79) | .716 | 1.08 | (0.72, 1.63) | 1 |
| **Text + Pictogram vs** | | | | | | | | | | | | |
| Text + Photograph | 0.89 | (0.62, 1.28) | 1 | 0.68 | (0.46, 1.01) | .385 | 0.85 | (0.52, 1.39) | 1 | 1.04 | (0.68, 1.57) | .871 |

*Note*. N = 1,755 (Immediately Post-Exposure) and N = 1,087 (Follow-up). P-values are adjusted using the Bonferroni-Holm method for multiple comparisons. Bolded p-values are significant at p < .05.

No HWL control; 2.14 in DrinkWise control) than among those in the high dose group (mean = 1.67 in No HWL control; 1.66 in DrinkWise control). Despite this, the pattern of differences between the intervention conditions and the two control conditions was the same in the low dose and high dose groups with just two exceptions. First, participants in the Text-Only condition had significantly greater negative emotional arousal than those in the No HWL condition only if they had a high dose of repeated exposure (Table 4). Second, those in the Text + Photograph condition had significantly greater negative emotional arousal than those in the Text-Only condition only if they had a high dose of repeated exposure (Table 4). This was due to a higher level of negative emotional arousal among those with a high dose (mean = 3.29) than low dose (mean = 2.97) of repeated exposure to the Text + Photograph HWLs (Fig 3).

There was a significant main effect of condition for positive emotional arousal (Table 2). Participants in all intervention conditions had lower positive emotional arousal than those in the No HWL control (Table 4). Participants in the Text + Photograph condition also had lower positive emotional arousal than those in the DrinkWise control (Table 4).

## Sensitivity analyses

Sensitivity analyses examined whether different cut-offs for the number of repeated exposure tasks completed (3+, 4+, 5+, 6+, 7+ and all 8) affected the pattern of results for the condition-by-dose interactions. S2 Table presents the proportion of participants who completed each cut-off value. S3 Table reports *p*-values for condition-by-dose interactions using each cut-off. As reported above there was one significant interaction effect for negative emotional arousal using the cut-off of 6+ as the reference. This remained statistically significant when using the cut-off of 3+, and the *p*-values for the cut-offs of 4+, 5+, 7+, and all 8 were all < .100. Lastly, there was a new significant interaction effect for past week alcohol consumption at follow-up when using a cut-off value of all 8 (*p* = .030). However, we did not investigate this interaction

**Table 4. Pairwise comparisons for negative emotional arousal and positive emotional arousal.**

| | Negative emotional arousal (Follow-up) | | | | Negative emotional arousal interaction effect (Follow-up) | | | | | | | | Positive emotional arousal (Follow-up) | | | |
| | | | | | Low Dose (0–5 repeated exposure tasks) | | | | High Dose (6–8 repeated exposure tasks) | | | | | | | |
| | β | (95% CI) | p | d | β | (95% CI) | p | d | β | 95% CI | p | d | β | (95% CI) | p | d |
|---|---|---|---|---|---|---|---|---|---|---|---|---|---|---|---|---|
| **No Health Warning Label vs** | | | | | | | | | | | | | | | | |
| Text-Only | 0.93 | (0.66, 1.19) | **< .001** | 0.42 | 0.58 | (0.09, 1.06) | .076 | 0.14 | 1.14 | (0.84, 1.45) | **< .001** | 0.45 | -0.55 | (-0.83, -0.26) | **< .001** | -0.23 |
| Text + Pictogram | 1.17 | (0.90, 1.43) | **< .001** | 0.52 | 0.81 | (0.32, 1.31) | **.006** | 0.20 | 1.38 | (1.07, 1.68) | **< .001** | 0.54 | -0.40 | (-0.68, -0.13) | **.024** | -0.17 |
| Text + Photograph | 1.33 | (1.06, 1.60) | **< .001** | 0.60 | 0.82 | (0.33, 1.32) | **.006** | 0.20 | 1.62 | (1.31, 1.93) | **< .001** | 0.63 | -0.72 | (-1.01, -0.43) | **< .001** | -0.29 |
| **DrinkWise vs** | | | | | | | | | | | | | | | | |
| Text-Only | 0.95 | (0.70, 1.19) | **< .001** | 0.46 | 0.59 | (0.15, 1.03) | **.045** | 0.16 | 1.15 | (0.86, 1.45) | **< .001** | 0.47 | -0.33 | (-0.61, -0.06) | .090 | -0.14 |
| Text + Pictogram | 1.18 | (0.93, 1.44) | **< .001** | 0.56 | 0.82 | (0.37, 1.28) | **< .001** | 0.22 | 1.39 | (1.09, 1.68) | **< .001** | 0.57 | -0.19 | (-0.46, 0.08) | .710 | -0.08 |
| Text + Photograph | 1.35 | (1.10, 1.60) | **< .001** | 0.64 | 0.83 | (0.38, 1.29) | **< .001** | 0.22 | 1.63 | (1.33, 1.92) | **< .001** | 0.66 | -0.51 | (-0.79, -0.22) | **.007** | -0.21 |
| **Text-Only vs** | | | | | | | | | | | | | | | | |
| Text + Pictogram | 0.24 | (-0.04, 0.52) | .190 | 0.10 | 0.24 | (-0.26, 0.73) | .698 | 0.06 | 0.24 | (-0.10, 0.58) | .346 | 0.08 | 0.14 | (-0.12, 0.41) | .578 | 0.06 |
| Text + Photograph | 0.40 | (0.12, 0.69) | **.015** | 0.17 | 0.25 | (-0.25, 0.74) | .981 | 0.06 | 0.47 | (0.13, 0.81) | **.021** | 0.17 | -0.17 | (-0.46, 0.11) | .693 | -0.07 |
| **Text + Pictogram vs** | | | | | | | | | | | | | | | | |
| Text + Photograph | 0.17 | (-0.12, 0.45) | .256 | 0.07 | 0.01 | (-0.50, 0.52) | .969 | 0.00 | 0.24 | (-0.11, 0.58) | .175 | 0.08 | -0.32 | (-0.60, -0.04) | .100 | -0.14 |

*Note.* N = 1,087 (Follow-up). P-values are adjusted using the Bonferroni-*Holm* method for multiple comparisons and d = Cohen's d effect size. Bolded p-values are significant at p < .05.

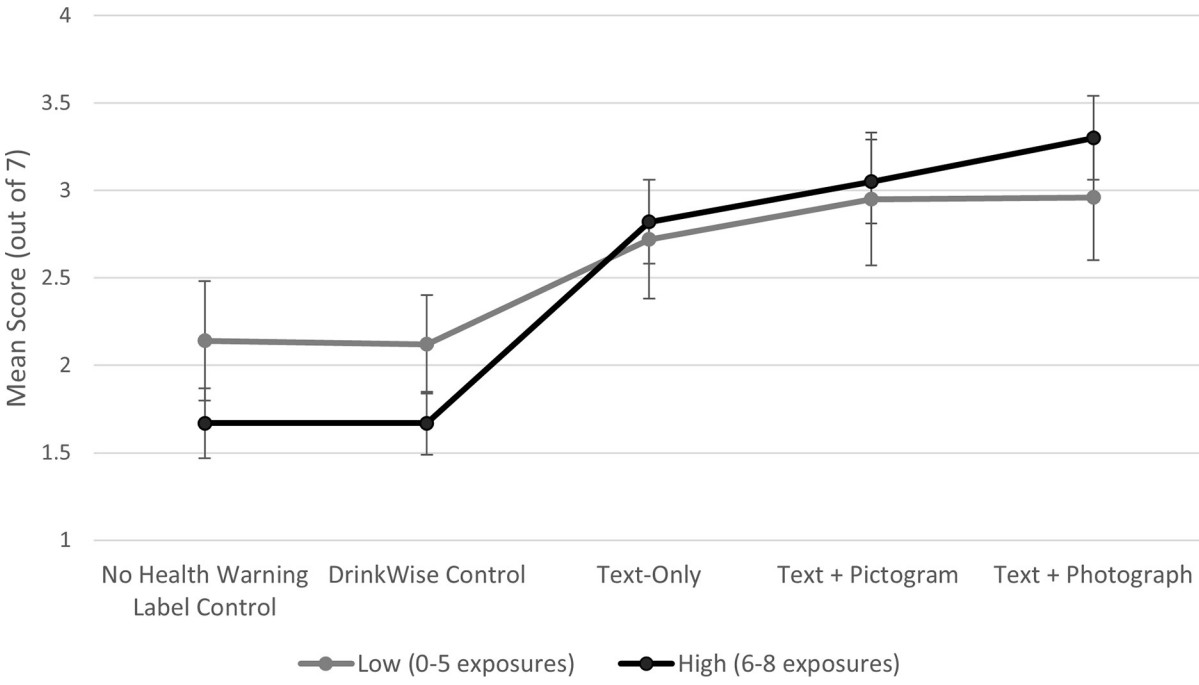

**Fig 3. Adjusted means for negative emotional arousal by condition type and dose of repeated exposure (*N = 1,087*).**

further, given the possibility it was due to chance as we conducted a large number of tests in these sensitivity analyses and this was the only significant interaction effect for (i) this outcome and (ii) this cut-off value. These results suggest that the number of repeated exposure tasks completed used to define a "high" dose of exposure was not likely to have had an impact on the main pattern of results.

## Discussion

This study examined the impact of exposure to new Text-Only, Text + Pictogram and Text + Photograph HWLs designed to communicate eight of the long-term harms associated with alcohol consumption. Compared to those who did not see any HWLs, participants in the Text + Pictogram condition were more likely to have intentions to avoid drinking alcohol completely in the next month (immediately post-exposure) and intentions to drink less alcohol in the next week (at follow-up), and participants in all three intervention conditions reported stronger negative emotional arousal and weaker positive emotional arousal. Compared to the DrinkWise label control participants, pairwise comparisons indicated that Text + Pictogram participants had stronger intentions to drink less alcohol in the next week and intentions to avoid drinking alcohol completely in the next month (at follow-up), while all three intervention conditions resulted in stronger negative emotional arousal and participants in the Text + Photograph condition also reported significantly weaker positive emotional arousal. This is despite the DrinkWise labels in our study being: slightly larger than they are in current practice; positioned on the front rather than back of alcohol containers; and in a consistent colour (dark text and pictograms for lightly coloured containers, light text and pictograms for dark coloured containers) rather than the varied brand-consistent colours that alcohol producers frequently use. These changes increased the comparability of the stimuli seen by participants across conditions, but also meant that the DrinkWise labels were more prominent and salient than they currently are. Nonetheless, the HWLs developed for this study still out-performed the 'enhanced' DrinkWise labels by eliciting negative emotional responses and reducing the excitement and pleasure drinkers felt when viewing alcohol container images. These findings are consistent with previous criticisms of the DrinkWise labels for vagueness and low emotional impact [17–20].

There was only one overall difference between the intervention conditions, with participants in the Text + Photograph condition reporting stronger negative emotional arousal than those in Text-Only condition. However, we also note the significant condition-by-dose interaction for negative emotional arousal. Overall, this significant interaction appeared to be driven by higher levels of negative emotional arousal among participants in the No HWL and DrinkWise control conditions who had a low dose rather than high dose of repeated exposure. A potential explanation for this pattern is that without effective HWLs, these images of alcohol containers effectively served as product advertising, with greater exposure undermining pre-existing moderate negative emotion levels. On the whole though, this made little difference to the impact of exposure to the intervention HWLs in comparison to the control conditions. There was also some indication that the Text + Photograph HWLs were more effective with a high dose compared to low dose of repeated exposure, such that only the high dose resulted in greater negative emotional arousal compared to those in the Text-Only condition. This may indicate that the richness of the information contained in Text + Photograph HWLs benefits from additional opportunities to engage with and process the full message. Additional studies that examine how the effects of different HWL formats vary over time are required to further explore this finding.

We found an overall main effect of the intervention conditions on awareness of increased cancer risk (but no significant differences in the pairwise comparisons), suggesting that the

four intervention HWLs that mentioned cancer provided new information. However, even in the intervention conditions, awareness of this link only reached approximately 70%, consistent with previous research demonstrating relatively lower awareness of the link between alcohol and cancer compared to other long-term harms [26]. By comparison, we noted potential ceiling effects for awareness of the link between alcohol and the risk of liver damage, heart disease and pregnancy complications (approximately 80–90% in all conditions). Indicating that these alcohol-related risks are already widely known, there was little capacity for exposure to the intervention HWLs to further increase awareness of liver damage and heart disease, and for the DrinkWise labels to further increase awareness of pregnancy complications. Nonetheless, despite these potential ceiling effects, it remains appropriate for these conditions to be included in future sets of alcohol HWLs to maintain their salience, help elicit negative emotional responses and dampen positive emotional responses.

Overall, these results provide evidence of potential benefits to public health if either of the three types of intervention HWLs—Text-Only, Text + Pictogram or Text + Photograph—were to be implemented. However, some results indicate that implementing Text + Pictogram HWLs should be preferred over Text-Only or Text + Photograph HWLs, given that they were the only HWLs to elicit stronger post-exposure intentions to avoid drinking completely in the next month and intentions to drink less in the next week at follow-up compared to the No HWL control, and stronger intentions to drink less in the next week at follow-up compared to the DrinkWise labels. Reviews of tobacco HWLs have consistently found pictorial warnings to be more effective than text-only warnings [30–33]. Although many of these studies have found that graphic (usually photographic) images are typically more effective than symbolic images [49, 64–67], these studies have typically been conducted in populations where text warnings had previously appeared on tobacco packs for many years, such that the graphic images added visual interest and emotional impact to familiar health information. By comparison, the current findings suggest that for populations who have not yet been routinely exposed to any alcohol HWLs, the most effective first step may be to require Text + Pictogram HWLs, rather than the more graphic and confronting Text + Photograph HWLs. For many jurisdictions, this may also be a more feasible first step for policy makers to take. Although, to reiterate, any of the intervention HWLs tested in this study would be an improvement over the current situation in most jurisdictions, where drinkers are exposed to either no HWL or poorly designed and relatively small text-only warnings about a limited range of topics [5]. They would also be an improvement over the current situation in Australia, where drinkers are exposed to the occasional low-impact DrinkWise label on the back of the container, or the situation from mid-2023, to a single text and pictogram pregnancy HWL that will have limited relevance for most drinkers [23].

Some aspects of the experimental design deserve noting. First, during the initial exposure session, intervention participants were exposed once to each of eight HWLs whereas DrinkWise participants were exposed twice to each of four labels, which may have increased the impact of the DrinkWise labels. This followed from our decision to use four DrinkWise current labels, so that our analyses compared possible future HWLs with the labels that Australians were sometimes viewing on containers. To make the DrinkWise condition more similar to our intervention conditions, the DrinkWise labels were also placed in a more prominent position than current DrinkWise labels, and they consistently appeared as either light coloured text/pictograms on a dark background or as dark coloured text/pictograms on a light background, rather than appearing in various brand-consistent colours. Overall, these factors increase confidence that our intervention HWLs are indeed more likely to be effective than the DrinkWise labels on Australian alcohol containers.

Second, the limited dose of exposure was insufficient to substantially change alcohol consumption in the week following initial exposure (controlling for past 7-day consumption at

baseline). For many drinkers, alcohol use is habitual and highly influenced by social and other external factors that may impede desired behavioural action [68–70]. It is promising that exposure to the Text + Pictogram warnings had effects on intentions to drink less and intentions to avoid alcohol completely, since intentions are an established, albeit imperfect, predictor of behaviour change [71]. Furthermore, the results indicated significant between-condition differences for negative emotional arousal, with all three intervention conditions leading to significantly higher scores than each control condition. This finding is notable, given previous evidence that feelings of displeasure mediated the effect of exposure to alcohol harm prevention advertisements on reduced urges to drink [72], and the evidence from tobacco control research that negative emotional arousal is a key pathway through which HWLs ultimately lead to intentions and behaviour change [73, 74]. Research using designs that achieve the correct temporal order between the mediator and outcome variables would be useful to test whether emotional responses mediate the effects of alcohol HWL exposure on subsequent changes in intentions and behaviours.

A key study strength is that the HWL stimuli were systematically developed through rigorous pre-testing that considered (i) awareness of short-term and long-term alcohol-related harms [75], (ii) the impact of small variations in the language used in the warnings and (iii) drinkers' ratings of the effectiveness of three potential pictograms and three potential photographs for each harm. Therefore, the specific warnings tested are strong exemplars of potential Text-Only, Text + Pictogram and Text + Photograph HWLs. However, one thing that these HWLs did not include is a reference to low-risk drinking guidelines. Recent research with alcohol HWLs [76] and alcohol harm prevention television advertisements [60, 77] has demonstrated that pairing messages communicating harms of alcohol with messages about recommended low-risk drinking guidelines can be particularly effective. Future research could investigate whether the effectiveness of the various HWLs tested in this study would be enhanced by inclusion of low-risk drinking guideline information.

An important study limitation is that participant recruitment was halted due to the emergence of COVID-19. High levels of concern about the pandemic may have eroded respondents' attention to and/or responsiveness to any HWL intervention. Just under 70% of the intended sample size was achieved, and the attrition rate at follow-up was larger than expected by the online panel, which may have reduced our power to detect differences between conditions on some outcomes. In addition, all pairwise comparisons were adjusted using the Bonferroni-Holm method to correct for multiple testing, further reducing the number of significant differences between conditions. A limitation of the experimental design is that exposure occurred online in an artificial scenario and involved viewing the HWLs on static images of alcohol containers. On the other hand, exposure to the stimuli in all conditions occurred in a consistent way via a series of Drink Choice Tasks, in which participants were instructed to choose between the different drinks on offer without any reference to the HWLs. This approach concealed the study purpose, thus minimising socially desirable responding. The ecological validity of exposure was increased by placing the HWLs on images of alcohol containers of the highest selling brands of participants' preferred type of alcohol, and by using repeated exposure tasks that better approximated routine exposure to HWLs as part of weekly activities. However, one week of repeated exposure may have been insufficient to produce meaningful effects in some outcomes. We also note that HWLs in the three intervention conditions were displayed in the same size on containers taking up around a quarter of the available space, which may have disadvantaged the more detailed images in the Text + Photograph condition. The effectiveness of these photographs may increase at larger sizes, for example, the 50% of the package used for tobacco warning labels in most countries [78].

Although participants were recruited from a non-probability panel, they came from a diverse range of sociodemographic backgrounds, and quotas for age groups matched the

distribution of adult weekly drinkers in a national benchmark study. However, some uncertainty remains as to whether effects would be replicated within the general population of drinkers. Finally, the sensitivity to change for some outcomes has not been psychometrically established, and our measure of alcohol consumption relied on self-report. In future studies, more accurate measurement of consumption using ecological momentary assessment methods could provide greater sensitivity for detecting even small changes [79], while field trials could provide critical real-world evidence for the impact of HWL implementation on population alcohol consumption [45].

## Conclusion

These findings add to growing evidence that HWLs on alcohol containers may have an important role to play as part of efforts to reduce alcohol-related harm [42, 45–47]. Alcohol HWLs are an intervention with population-wide reach that provide consumers with accurate information about the health harms, and they provide this information at the point of purchase and consumption. Any of the three types of HWLs tested in this study would be an improvement over the current situation in many jurisdictions, although there is some evidence that implementing Text + Pictogram HWLs should be recommended over Text-Only or Text + Photograph warnings.

## Supporting information

**S1 Fig. Example of Drink Choice Task brand and warning label type randomisation, by condition type.**
(TIF)

**S2 Fig. Example of how the Drink Choice Tasks appeared to participants, by condition type.**
(TIF)

**S3 Fig. Intervention alcohol warning labels by condition.** Note: We had initially pre-tested the harm topic 'alcohol increases your risk of 8 different types of cancer'. After careful consideration and further review of the body of evidence on the causal relationship between alcohol and cancer, we changed '8 types of cancer' to '7 types of cancer' and removed stomach cancer from the list of cancers. While there is evidence to suggest a strong causal link between alcohol and stomach cancer, the evidence is probable and not yet convincing.
(TIF)

**S1 Table. Sample characteristics at follow-up (N = 1,087).** Note. HWL = health warning label; SD = standard deviation; SE = standard error; NHMRC = National Health and Medical Research Council; LTH = long-term harm; STH = short-term harm.
(PDF)

**S2 Table. Sensitivity analyses: Frequencies and percentages of participants who completed the repeat exposure tasks for the difference cut-offs at follow-up (N = 1,087).** Note: '6 or more' is listed first as it is the cut-off point we reported in our results.
(PDF)

**S3 Table. Omnibus test results for the condition-by-dose interaction term for different cut-offs for the completed number of repeat exposure tasks.** Note. p-value for condition × dose interaction shown; and † = HC3 heteroskedasticity-consistent standard error estimator used. '6 or more' is listed first as it is the cut-off point we reported in our results.
(PDF)

**S1 Appendix. Regression analyses output for Tables 2–4 and Fig 3.**
(PDF)

## Author Contributions

**Conceptualization:** Emily Brennan, Kimberley Dunstone, Sarah Durkin, Michael D. Slater, Janet Hoek, Simone Pettigrew, Melanie Wakefield.

**Formal analysis:** Amanda Vittiglia, Sam Mancuso.

**Funding acquisition:** Emily Brennan, Sarah Durkin, Michael D. Slater, Janet Hoek, Simone Pettigrew, Melanie Wakefield.

**Methodology:** Emily Brennan, Kimberley Dunstone, Amanda Vittiglia, Sarah Durkin, Michael D. Slater, Janet Hoek, Simone Pettigrew, Melanie Wakefield.

**Project administration:** Kimberley Dunstone, Amanda Vittiglia.

**Supervision:** Emily Brennan, Melanie Wakefield.

**Visualization:** Emily Brennan, Amanda Vittiglia, Sam Mancuso.

**Writing – original draft:** Emily Brennan, Amanda Vittiglia, Sam Mancuso.

**Writing – review & editing:** Kimberley Dunstone, Sarah Durkin, Michael D. Slater, Janet Hoek, Simone Pettigrew, Melanie Wakefield.

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
