## [Decision Letter · Decision Letter 0]

15 Jun 2022

PONE-D-21-36330Testing the effectiveness of alcohol health warning label formats: An online experimental study with Australian adult drinkersPLOS ONE

Dear Dr. Brennan,

Thank you for submitting your manuscript to PLOS ONE. After careful consideration, we feel that it has merit but does not fully meet PLOS ONE’s publication criteria as it currently stands. Therefore, we invite you to submit a revised version of the manuscript that addresses the points raised during the review process.

We look forward to receiving your revised manuscript.

Kind regards,

Rafael Aiello Bomfim, Phd

Academic Editor

PLOS ONE

**Journal requirements:**

a) Did participants provide their written or verbal informed consent to participate in this study?

“I have read the journal’s policy and the authors of this manuscript have the following competing interests: EB, KD, AV, SD and MW are employed by a non-profit organisation that conducts research, public health interventions and advocacy aimed at reducing alcohol-related health harms in the community, especially those pertaining to cancer. EB, SP and MW have received other NHMRC grants on alcohol harm communication. EB is a member of the Expert Reference Group for the Alcohol.Think Again campaign. JH has provided advice to Alcohol Healthwatch, a New Zealand non-profit organisation that conducts advocacy to reduce alcohol-related health harms in the community. SP is a Board Member of the Foundation for Alcohol Research and Education, and a member of government advisory committees on alcohol.”

**Additional Editor Comments:**

there are some points that deserve attention as pointed out by the reviewer.

Reviewers' comments:

Reviewer's Responses to Questions

**Comments to the Author**

1. Is the manuscript technically sound, and do the data support the conclusions?

Reviewer #1: Yes

2. Has the statistical analysis been performed appropriately and rigorously? 

Reviewer #1: Yes

3. Have the authors made all data underlying the findings in their manuscript fully available?

Reviewer #1: Yes

4. Is the manuscript presented in an intelligible fashion and written in standard English?

Reviewer #1: Yes

5. Review Comments to the Author

Reviewer #1: This study aims to experimentally test the effects of alcohol container health warning labels varying by label format and health message versus control labels with no or an industry-developed warning among Australian adult alcohol consumers. This is a well-designed and carefully executed RCT, I particularly applaud the innovative approach for testing label effects immediately following label exposure as well as after 8 days of label exposure. Minor suggestions and questions to consider are listed below.

Introduction

Line 70: I am not familiar with the term “textual”, would you consider “weak message content, and most lack images”?

Line 71: Is there value in adding an example of a situation harm that allow most drinkers to self-exempt (e.g., drinking when pregnant)?

Line 71: Please note that South Korea mandates 1 or 3 HWL on alcohol containers, including 1 label with a cancer warning. However, industry selects which 1 of the 3 labels to use. Moreover, although Ireland has passed legislation mandated HWL with a cancer message, it has not yet been implemented due to a pause caused by industry pushback. At a minimum, please add a statement about the unprecedented pause in implementing the label regulations in Ireland.

Line 75: How long has DrinkWise Australia been voluntarily using labels on alcohol containers? Was this a new initiative in 2020?

Line 78: Please add: “a pregnancy HWL on all alcohol containers, with a minimum size requirement and a white background….”.

Line 85: Given that any amount of alcohol increases cancer risk, what is the definition of “risky alcohol consumption”? I suggest revising this statement by removing “risky”.

Line 88: Is this 50% of the front of the package or 50% of the entire tobacco package?

Line 88: Please add “full colour” before “image and covering at least 50% of the package”. Is it also worth highlighting the efficacy of rotating messages on tobacco HWL to reduce message wear out?

Line 102: Please update this statement to read: “educating about the number of standard drinks in a container”.

Line 102: The labels in the Yukon study were not placed on the front of the container – but somewhere on the container to avoid overlapping with industry labels.

Line 103: The usual practice at the 2 control liquor stores included a long-standing label warning about drinking when pregnant, when operating a car or machinery, and general health risks.

Lines 104 – 107: As described in the scoping review by Kokole et al., 2021, the Yukon labelling study was designed based on the theory that exposure to well-designed labels over time can increase attention to and deep cognitive processing of label information, leading to increased knowledge of alcohol-related harms and behaviour change. Whereas most lab-based labelling studies are largely focused on an immediate reaction after a one-time exposure to labels.

Line 109: This sentence is awkward, please revise.

Pictorial, image, and photograph are used interchangeably throughout the Introduction. This is ok but I wanted to raise to ensure it is intentional.

Lines 114-115: Revise to “a range of relatively lesser known alcohol-related chronic health harms”.

Methods

Lines 126-128: Use #s instead of letters to list the 5 conditions.

Lines 128-129: Revise to: “In each of the three intervention label conditions, participants were exposed to eight HWLs depicting a different alcohol-related health harm.”

Lines 133-136: Why was power calculation based on the outcome, intentions to limit to drinking? What was the 30% attrition rate based on?

Line 156: Is 18 years the minimum legal drinking age in Australia? If yes, please indicate so.

Lines 176-177: Was the order participants were exposed to the 8 HWLs randomized? Why were the 8 HWLs selected? Why 8?

Line 207: What text message was featured on the HWL for each harm? Please provide in the main text of the paper.

Line 312: If participants did not open any of the emails in the follow-up period but completed the follow survey, they were still included in the “low dose” group in the analyses? Why was 6 days selected as the cut off? Would participants exposed to 0 image emails in follow be different than participants who were exposed to 5 image emails? Was this decision driven by the data? I now see this is tested in the sensitivity analyses, but I wonder if a very brief explanation could be provided earlier?

Discussion

Line 442: Please provide an explanation and describe the research and policy implications for high dose exposure resulting in greater negative emotional arousal for the Text + Photograph HWL vs Text-only HWL condition. For example, consistent with evidence of effective labels for both tobacco and alcohol, the photograph HWL may draw consumer attention to key messages and support greater cognitive processing over time. This is important to consider as deep engagement with labels has been shown to be the strongest predictor of motivating consumers to change their behaviour. Therefore, it is critical for more evidence including a post-follow up component as you have in your study (vs. only including an immediate response to labels). I have provided below examples of references to consider.

Brewer et al. (2019). Understanding why pictorial pack warnings increase quit attempts. Annals of Behavioral Medicine, 53:232-243.

Hammond, D. (2011). Health warning messages on tobacco products: a review. Tobacco Control, 20:327-337.

Hobin et al. (2020). Effects of strengthening alcohol labels on attention, message processing, and perceived effectiveness: A quasi-experimental study in Yukon, Canada. International Journal of Drug Policy, 77.

Lines 470-471: Please add a statement about current labels being poorly designed in terms of position on container, size, use of colour or border, etc. For example, please consider: “…, where drinkers are exposed to either no HWL or a poorly designed, text-only warning about a limited range of topics that is relatively small in size.”

A final suggestion for the discussion is to highlight evidence demonstrating the effectiveness of health warning messages (‘why to reduce’) in combination with drinking guidelines (how to reduce) to encourage drinkers to attempt to reduce their alcohol consumption, and encourage future research to test this label message combination.

6. PLOS authors have the option to publish the peer review history of their article (what does this mean?). If published, this will include your full peer review and any attached files.

Reviewer #1: No

---

## [Author Response · Author response to Decision Letter 0]

7 Aug 2022

Response to Reviewers

Response: We have ensured that our manuscript meets PLOS ONE’s style requirements.

a) Did participants provide their written or verbal informed consent to participate in this study?

Response: Response: We have amended our ethics statement to specify that when participants implied their consent to participate by clicking through to participate in the study, this was collected in place of formal written or verbal consent. This procedure was approval by our ethics committee:

Ethical approval for the study was obtained from Cancer Council Victoria’s Institutional Research Review Committee (IER 1609). Participants received an email invitation to participate in the study, clicked through to the study, were provided with information about the broad aims of the study, completed screening questions and then implied consent by clicking through to further questions. This implied consent was collected in place of formal written or verbal consent. This procedure was approved by the Institutional Research Review Committee.

“I have read the journal’s policy and the authors of this manuscript have the following competing interests: EB, KD, AV, SD and MW are employed by a non-profit organisation that conducts research, public health interventions and advocacy aimed at reducing alcohol-related health harms in the community, especially those pertaining to cancer. EB, SP and MW have received other NHMRC grants on alcohol harm communication. EB is a member of the Expert Reference Group for the Alcohol.Think Again campaign. JH has provided advice to Alcohol Healthwatch, a New Zealand non-profit organisation that conducts advocacy to reduce alcohol-related health harms in the community. SP is a Board Member of the Foundation for Alcohol Research and Education, and a member of government advisory committees on alcohol.”

Response: We have revised our Competing Interests statement and updated it as directed. This has been included this in the online submission form/cover letter :

“I have read the journal’s policy and the authors of this manuscript have the following competing interests: EB, KD, AV, SD and MW are employed by a non-profit organisation that conducts research, public health interventions and advocacy aimed at reducing alcohol-related health harms in the community, especially those pertaining to cancer. EB, SP and MW have received other NHMRC grants on alcohol harm communication. MS has current grant support for alcohol-related research from the NIH National Institute on Drug Abuse and has previously received grants from the National Institute on Alcohol Abuse and Alcoholism, National Cancer Institute, and the Robert Woods Johnson Foundation. EB and SP are members of the Expert Reference Group for the Alcohol. Think Again campaign. JH has provided advice to Alcohol Healthwatch, a New Zealand non-profit organisation that conducts advocacy to reduce alcohol-related health harms in the community. This does not alter our adherence to PLOS ONE policies on sharing data and materials.”

Response: We acknowledge that publication of the manuscript will not proceed until we have provided the relevant accession numbers or DOIs necessary for our data to be accessed. We do not require any changes to be made to our Data Availability statement.

Additional Editor Comments:

There are some points that deserve attention as pointed out by the reviewer.

Response: As detailed below, we have addressed each of the reviewer’s comments.

Reviewers' Comments:

1. This study aims to experimentally test the effects of alcohol container health warning labels varying by label format and health message versus control labels with no or an industry-developed warning among Australian adult alcohol consumers. This is a well-designed and carefully executed RCT, I particularly applaud the innovative approach for testing label effects immediately following label exposure as well as after 8 days of label exposure. Minor suggestions and questions to consider are listed below.

Response: Thank you for this positive feedback, and for your thoughtful review. We have addressed each of your comments and questions below and believe the manuscript has further improved because of these changes.

Introduction

2. Line 70: I am not familiar with the term “textual”, would you consider “weak message content, and most lack images”?

Response: We have revised this sentence to read “…weak written messages[6] and most lack images,[7] have low noticeability…”.

3. Line 71: Is there value in adding an example of a situation harm that allow most drinkers to self-exempt (e.g., drinking when pregnant)?

Response: We have revised this sentence to read “…and feature harms that allow most drinkers to self-exempt (e.g., drinking when pregnant)”.

4. Line 71: Please note that South Korea mandates 1 or 3 HWL on alcohol containers, including 1 label with a cancer warning. However, industry selects which 1 of the 3 labels to use. Moreover, although Ireland has passed legislation mandated HWL with a cancer message, it has not yet been implemented due to a pause caused by industry pushback. At a minimum, please add a statement about the unprecedented pause in implementing the label regulations in Ireland.

Response: Thank you for this suggestion. We have revised this sentence to contain more detail about the situation in South Korea and Ireland: “To date, HWLs that warn about the link between alcohol and cancer have been mandated in South Korea (although industry can select which one of three HWLs is placed on the container and only one of the three mentions cancer)[9] and in Ireland,[10] although industry interference there has reportedly delayed both the passage of the bill and the introduction of the HWLs.[11-13]” 

5. Line 75: How long has DrinkWise Australia been voluntarily using labels on alcohol containers? Was this a new initiative in 2020?

Response: This sentence has been revised to specify that DrinkWise “…has produced labels since 2011…”.

6. Line 78: Please add: “a pregnancy HWL on all alcohol containers, with a minimum size requirement and a white background….”.

Response: We have added “…to appear on all alcohol containers…” to this sentence.

7. Line 85: Given that any amount of alcohol increases cancer risk, what is the definition of “risky alcohol consumption”? I suggest revising this statement by removing “risky”.

Response: We have removed “risky” from this sentence. 

8. Line 88: Is this 50% of the front of the package or 50% of the entire tobacco package?

Response: We have specified that this is “…50% of the front of the package.”

9. Line 88: Please add “full colour” before “image and covering at least 50% of the package”. Is it also worth highlighting the efficacy of rotating messages on tobacco HWL to reduce message wear out?

Response: We have added “full colour” before “image and covering at least…”. We have also added a sentence to acknowledge the potential for HWL effects to wear out over time and the benefit of using multiple HWLs at once to help mitigate such effects (while acknowledging even that isn’t completely effective, such that there needs to be frequent introduction of new HWLs):

“The effects of tobacco HWLs have been found to wear-out over time, as smokers stop paying attention to them.[35-37] To slow down these wear-out effects, many tobacco HWL policies require a set of HWLs to be implemented and/or for two or more sets of HWLs to be rotated (although more recent research indicates that these measures are not completely effective at preventing wear-out such that frequent introduction of new HWLs is required).[38]” 

10. Line 102: Please update this statement to read: “educating about the number of standard drinks in a container”.

Response: This sentence has been revised as suggested.

11. Line 102: The labels in the Yukon study were not placed on the front of the container – but somewhere on the container to avoid overlapping with industry labels.

Response: Thank you for pointing this out; we have removed “…the front of…” from this sentence.

12. Line 103: The usual practice at the 2 control liquor stores included a long-standing label warning about drinking when pregnant, when operating a car or machinery, and general health risks.

Response: Thank you – we have added this detail: “…while usual practice (i.e., long-standing warnings about drinking when pregnant, and about the impact of drinking on operating a car or machinery and general health risks[45])…”.

13. Lines 104 – 107: As described in the scoping review by Kokole et al., 2021, the Yukon labelling study was designed based on the theory that exposure to well-designed labels over time can increase attention to and deep cognitive processing of label information, leading to increased knowledge of alcohol-related harms and behaviour change. Whereas most lab-based labelling studies are largely focused on an immediate reaction after a one-time exposure to labels.

Response: Thank you for pointing this out. However, we don’t believe this comment requires any changes to the manuscript.

14. Line 109: This sentence is awkward, please revise.

Response: We have revised this sentence from: “Initial implementation options for alcohol HWLs may be prominent text-only warnings or pictorial warnings that feature simple stylised pictograms or that feature graphic photographs.” 

to: “Jurisdictions that are implementing alcohol HWLs for the first time will need to decide whether to implement prominent text-only warnings, or pictorial warnings that feature simple stylised pictograms or that feature graphic photographs.”

15. Pictorial, image, and photograph are used interchangeably throughout the Introduction. This is ok but I wanted to raise to ensure it is intentional.

Response: We have used “pictorial” when referring to this type of HWL in a general way (i.e., those that include an image), “image” when referring more specifically to the content or design of particular HWLs or HWL policies, and “photograph” when referring specifically to those that include a photograph (as opposed to some other type of image, such as a pictogram or diagram or illustration). 

16. Lines 114-115: Revise to “a range of relatively lesser known alcohol-related chronic health harms”.

Response: We have revised this sentence to read “Second, our suite of systematically developed alcohol HWLs covers a range of relatively lesser known alcohol-related long-term health effects.”

Methods

17. Lines 126-128: Use #s instead of letters to list the 5 conditions.

Response: We have made this change.

18. Lines 128-129: Revise to: “In each of the three intervention label conditions, participants were exposed to eight HWLs depicting a different alcohol-related health harm.”

Response: We have made the recommended change to the start of the sentence but have kept the “each depicting a different alcohol-related harm”, as this helps to make it clearer that eight different harms were depicted across the eight different HWLs.

19. Lines 133-136: Why was power calculation based on the outcome, intentions to limit to drinking? What was the 30% attrition rate based on?

Response: The power calculation was based on the “intentions to limit drinking at follow-up” variable as we had access to data from a similar study with a similar outcome measure that could be used as the basis of these power calculations. By comparison, we did not have access to similar data regarding changes in past week alcohol consumption at follow-up. Intentions are a well-established (albeit imperfect) predictor of subsequent behaviour change and are commonly used as outcomes in experimental tests of message effectiveness. 

The expected 30% attrition rate was provided by the online non-probability panel used to recruit participants and was based on their prior experience conducting similar studies. We have added this detail in parentheses at the end of the sentence: “(anticipated by the online panel provider)”.

20. Line 156: Is 18 years the minimum legal drinking age in Australia? If yes, please indicate so.

Response: Yes. We have added this detail in parentheses at the end of the sentence: “(18 is the minimum legal drinking age in Australia).”

21. Lines 176-177: Was the order participants were exposed to the 8 HWLs randomized? Why were the 8 HWLs selected? Why 8?

Response: Yes, the order in which participants were exposed to the 8 HWLs was randomised across participants – we have clarified this detail in the text. We have also added a sentence to the end of this paragraph to explain that we chose to use 8 HWLs to strike a balance between covering a range of health effects and ensuring sufficient exposure to the HWLs (i.e., why 8 rather than a smaller number) without overburdening participants (i.e., why 8 rather than a larger number). 

22. Line 207: What text message was featured on the HWL for each harm? Please provide in the main text of the paper.

Response: We have revised this paragraph so that rather than listing the eight harms, we provide the text message featured on the HWLs for each harm.

23. Line 312: If participants did not open any of the emails in the follow-up period but completed the follow survey, they were still included in the “low dose” group in the analyses? Why was 6 days selected as the cut off? Would participants exposed to 0 image emails in follow be different than participants who were exposed to 5 image emails? Was this decision driven by the data? I now see this is tested in the sensitivity analyses, but I wonder if a very brief explanation could be provided earlier?

Response: In total, only 10% of participants did not complete any of the repeat exposure tasks but still completed the follow-up survey. We have made changes to two sections of the manuscript to further explain why these participants were included in the analyses, and to justify our decision to dichotomise the number of repeat exposure tasks at six or more. 

In Paragraph 4 of the Procedure section, we have added the following sentence after explaining that participants could choose to complete the repeated exposure tasks on some days but not others:

“By allowing variation in the number of repeated exposure tasks completed by each participant (between zero and eight), we aimed to mimic the real-life scenario whereby consumers are exposed to alcohol containers on a regular basis, but with a frequency that varies across consumers and over time (i.e., even a regular drinker will be exposed more frequently on some weeks than others).”

We have also revised Paragraph 3 of the Statistical Analysis section to explain that:

“For the primary analyses, a “high dose” of exposure was defined as having completed at least six of the eight repeat exposure tasks and a “low dose” of exposure was defined as having completed zero to five repeat exposure tasks; however, sensitivity analyses explored whether the same pattern of effects was observed when using different cut-offs. Defining a “high dose” of exposure as comprising six or more tasks recognised that although participants were incentivised to complete all eight repeat exposure tasks, it was unrealistic to expect that most participants would be able to comply with this demand of the study. Setting the cut-point at six or more captured those who completed at least the majority (>75%) of repeat exposure tasks. Including those who did not complete any repeat exposure tasks in the “low dose” category recognises that in the real world, in any given week, there would be some drinkers who were not exposed to any alcohol HWLs, such that no repeat exposure is part of the expected natural variation.”

Discussion

24. Line 442: Please provide an explanation and describe the research and policy implications for high dose exposure resulting in greater negative emotional arousal for the Text + Photograph HWL vs Text-only HWL condition. For example, consistent with evidence of effective labels for both tobacco and alcohol, the photograph HWL may draw consumer attention to key messages and support greater cognitive processing over time. This is important to consider as deep engagement with labels has been shown to be the strongest predictor of motivating consumers to change their behaviour. Therefore, it is critical for more evidence including a post-follow up component as you have in your study (vs. only including an immediate response to labels). I have provided below examples of references to consider.

Brewer et al. (2019). Understanding why pictorial pack warnings increase quit attempts. Annals of Behavioral Medicine, 53:232-243.

Hammond, D. (2011). Health warning messages on tobacco products: a review. Tobacco Control, 20:327-337.

Hobin et al. (2020). Effects of strengthening alcohol labels on attention, message processing, and perceived effectiveness: A quasi-experimental study in Yukon, Canada. International Journal of Drug Policy, 77.

Response: Each of the recommended references are useful for understanding how warnings work to change behaviours. However, because this effect of the high dose versus low dose of exposure to the Text + Photograph HWLs was only observed for one outcome (negative emotional arousal) and compared to only one other condition (Text-Only, but neither of the control conditions), we are reluctant to overinterpret this single finding. However, we have added the following two sentences to the end of Paragraph 2 in the Discussion where we describe this finding: “This may indicate that the richness of the information contained in Text + Photograph HWLs benefits from additional opportunities to engage with and process the full message. Additional studies that examine how the effects of different HWL formats vary over time are required to further explore this finding.”

25. Lines 470-471: Please add a statement about current labels being poorly designed in terms of position on container, size, use of colour or border, etc. For example, please consider: “…, where drinkers are exposed to either no HWL or a poorly designed, text-only warning about a limited range of topics that is relatively small in size.”

Response: We have revised this sentence to read: “…where drinkers are exposed to either no HWL or poorly designed and relatively small text-only warnings about a limited range of topics[5].”

26. A final suggestion for the discussion is to highlight evidence demonstrating the effectiveness of health warning messages (‘why to reduce’) in combination with drinking guidelines (how to reduce) to encourage drinkers to attempt to reduce their alcohol consumption, and encourage future research to test this label message combination.

Response: Thank you for this suggestion. In Paragraph 7 of the Discussion, we have added the following three sentences: “However, one thing that these HWLs did not include is a reference to low-risk drinking guidelines. Recent research with alcohol HWLs[76] and alcohol harm prevention television advertisements[60, 77] has demonstrated that pairing messages communicating harms of alcohol with messages about recommended low-risk drinking guidelines can be particularly effective. Future research could investigate whether the effectiveness of the various HWLs tested in this study would be enhanced by inclusion of low-risk drinking guideline information.”

---

## [Decision Letter · Decision Letter 1]

2 Oct 2022

Testing the effectiveness of alcohol health warning label formats: An online experimental study with Australian adult drinkers

PONE-D-21-36330R1

Dear Dr. Brennan,

We’re pleased to inform you that your manuscript has been judged scientifically suitable for publication and will be formally accepted for publication once it meets all outstanding technical requirements.

Kind regards,

Rafael Aiello Bomfim, Phd

Academic Editor

PLOS ONE

Additional Editor Comments (optional):

Congratulations

Reviewers' comments:

Reviewer's Responses to Questions

**Comments to the Author**

1. If the authors have adequately addressed your comments raised in a previous round of review and you feel that this manuscript is now acceptable for publication, you may indicate that here to bypass the “Comments to the Author” section, enter your conflict of interest statement in the “Confidential to Editor” section, and submit your "Accept" recommendation.

Reviewer #1: All comments have been addressed

2. Is the manuscript technically sound, and do the data support the conclusions?

Reviewer #1: Yes

3. Has the statistical analysis been performed appropriately and rigorously? 

Reviewer #1: Yes

4. Have the authors made all data underlying the findings in their manuscript fully available?

Reviewer #1: Yes

5. Is the manuscript presented in an intelligible fashion and written in standard English?

Reviewer #1: Yes

6. Review Comments to the Author

Reviewer #1: (No Response)

7. PLOS authors have the option to publish the peer review history of their article (what does this mean?). If published, this will include your full peer review and any attached files.

Reviewer #1: No
